# *MoNOT3* Subunit Has Important Roles in Infection-Related Development and Stress Responses in *Magnaporthe oryzae*

**DOI:** 10.3390/ijms25063290

**Published:** 2024-03-14

**Authors:** Youngmin Kim, Miju Jo, Sunmin An, Yerim Lee, Eu Ddeum Choi, Min-Hye Jeong, Ki-Tae Kim, Sook-Young Park

**Affiliations:** 1Department of Plant Medicine, Sunchon National University, Suncheon 57922, Republic of Korea; ymkim102@korea.kr (Y.K.); miju7188@gmail.com (M.J.); sminan201@gmail.com (S.A.); yelee117@gmail.com (Y.L.); nayaced@korea.kr (E.D.C.); minhye1962@gmail.com (M.-H.J.); kitaekim@scnu.ac.kr (K.-T.K.); 2Fruit Research Institute, Jellanamdo Agricultural Research and Extension Servieces, Haenam 59021, Republic of Korea; 3Interdisciplinary Program in IT-Bio Convergence System (BK21 Plus), Sunchon National University, Suncheon 57922, Republic of Korea

**Keywords:** appressorium, CCR4-NOT complex, conidiation, functional analysis, *Magnaporthe oryzae*, pathogenicity

## Abstract

The multifunctional carbon catabolite repression negative on TATA-box-less complex (CCR4-NOT) is a multi-subunit complex present in all eukaryotes, including fungi. This complex plays an essential role in gene expression; however, a functional study of the CCR4-NOT complex in the rice blast fungus *Magnaporthe oryzae* has not been conducted. Seven genes encoding the putative CCR4-NOT complex were identified in the *M. oryzae* genome. Among these, a homologous gene, *MoNOT3*, was overexpressed during appressorium development in a previous study. Deletion of *MoNOT3* in *M. oryzae* resulted in a significant reduction in hyphal growth, conidiation, abnormal septation in conidia, conidial germination, and appressorium formation compared to the wild-type. Transcriptional analyses suggest that the *MoNOT3* gene affects conidiation and conidial morphology by regulating *COS1* and *COM1* in *M. oryzae*. Furthermore, *Δmonot3* exhibited a lack of pathogenicity, both with and without wounding, which is attributable to deficiencies in the development of invasive growth *in planta*. This result was also observed in onion epidermal cells, which are non-host plants. In addition, the *MoNOT3* gene was involved in cell wall stress responses and heat shock. Taken together, these observations suggest that the *MoNOT3* gene is required for fungal infection-related cell development and stress responses in *M. oryzae*.

## 1. Introduction

While the canonical TATAAA sequence is a prevalent core promoter element in eukaryotes, not all core promoters contain this sequence, leading to the identification of “TATA-less” promoters [1]. One such regulator is the carbon-catabolite repression-negative on TATA-box-less (CCR4-Not) complex. The CCR4-Not complex is a highly conserved multi-subunit protein found in eukaryotic cells, including fungi, and plays an essential role in regulating gene expression. These functions include interacting with transcription factors, initiating transcription, elongating transcription, exporting messenger RNA (mRNA), degrading mRNA, deadenylating mRNA, and repressing mRNA translation. The complex is consistently found in the nucleus and cytoplasm and is present at all stages of gene expression throughout the life cycle, indicating its importance [2,3,4].

In yeast, a classic model fungus, the CCR4-Not complex is structurally composed of nine conserved subunits, including CCR4, associated factor *Caf* genes (*Caf130*, *Caf40*, and *Caf1*), as well as five *Not* genes (*Not1*, *Not2*, *Not3* (*SIG1*), *Not4* (*Mot2*), *Not5*) [2,4]. Through the genetic analysis of yeast mutants, the CCR4 protein was identified as a positive regulator of glucose-repressible alcohol dehydrogenase (*ADH2*) genes [5], and the *CCR4*-associated factor (*Caf*) gene was identified as a catalytic subunit [2,6].

NOT1 proteins act as scaffolds for assembling the complex, providing binding sites for the NOT2-NOT3/5 subunits, CAF130, CAF40, and the catalytic module [7]. However, the precise function of NOT3 in yeast remains unclear. In humans, *CNOT3* has been implicated in gene expression, regulation, mRNA surveillance, and export. This suggests that this gene plays a crucial role in post-transcriptional gene regulation and is essential for normal cell function and viability [8]. In most eukaryotes, except for yeast and *Candida albicans*, *NOT3* genes are orthologous to yeast *NOT3* and *NOT5*. In human fungal pathogens, such as *Cryptococcus neoformans* and *Candida albicans*, *CaNOT5* is involved in morphogenesis, virulence, and cell wall structure [9,10].

Similarly, in plant pathogenic fungi, the CCR4-NOT complex has been found to play a crucial role in regulating virulence-related functions. In the case of *Fusarium oxysporum* f. sp. *niveum*, which causes *Fusarium* wilt in watermelon, the *FoNOT2* gene plays diverse roles, including mycelial growth, conidiation, virulence, cell wall integrity, and the production of secondary metabolites [11]. *Fusarium graminearum*, the pathogen responsible for *Fusarium* head blight in wheat, has also been studied previously. *FgNOT3* is involved in pleiotropic defects in vegetative growth, sexual reproduction, secondary metabolite production, and virulence [12]. However, to date, there have been no reports on the gene functional analysis of the NOT complex, including *NOT3*, in *Magnaporthe oryzae*, which causes rice blast disease.

Rice blast caused by *M. oryzae* is one of the most destructive diseases affecting cultivated rice worldwide. *M. oryzae* has been used as a model to understand fungal diseases [13]. To infect a host cell, *M. oryzae* undergoes several developmental processes, including attachment, conidial germination, appressorium formation, direct penetration using turgor pressure, and invasive growth into the rice host cell. As an initial step in invading a host plant, three-celled asexual conidia attach to rice leaves after landing on the host surface [14]. When conidia are attached to a leaf with mucilage, they germinate within 3–4 h and form an appressorium within 4–6 h. A specialised structure called an appressorium directly penetrates the host cell with an enormous internal turgor pressure, which is estimated to be as high as 8.0 MPa [15]. Several studies have reported that signalling-related pathways are involved in appressorium formation, including the cAMP-related pathway [16], MAP kinase pathway [17], and calcium/calmodulin-dependent signalling [18,19]. Genes related to signalling regulate the formation of appressoria, as well as penetration and invasive growth [15,20].

Invasive hyphae colonise host cells and express numerous genes associated with pathogenicity [21]. Subsequently, conidia of *M. oryzae* emerge on the infected rice cells to reinitiate the disease cycle. Conidia produced from conidiophores are controlled by several genes, including *Com1*, *Con7*, and *Con1* [20,22,23]. Developmental processes, such as conidiation, conidial attachment, germination, appressorium formation, and pathogenicity, are regulated by genes, including transcription factors and protein complexes [24,25,26,27]. However, the pathosystem mechanisms between the host cells and *M. oryzae* need to be elucidated. Understanding the genetic mechanisms underlying host infections is a priority for the development of new disease-control strategies.

In a previous study, we identified a gene, MGG_08101, which was specifically abundant during appressorium development compared to mycelial growth in *M. oryzae* [28]. The gene encoding putative *NOT3/5* was named *MoNOT3*. In this study, we conducted a functional analysis of and *MoNOT3* in *M. oryzae*. To analyse the function of this gene under infection-related conditions, we generated *MoNOT3* deletion mutants. Our data show that *MoNOT3* is necessary for the overall fungal development of *M*. *oryzae* infection, particularly for pathogenicity and thermal stress response.

## 2. Results

### 2.1. Molecular Characterisation of MoNOT3

The CCR4-NOT complex consists of CCR4 and NOT proteins in eukaryotes, except for *Saccharomycetaceae* and *Candida albicans*, which contain additional Caf protein subunits. Members of *NOT3/5* possess conserved components, including the CCR4-NOT complex, N-terminal (IPR007207), *NOT2*, *NOT3*, and *NOT5* (IPR007282). In a previous study, *MGG_08101* was upregulated during appressorium development. This gene was identified as an ortholog of *NOT3/5* in *Saccharomyces cerevisiae* using a BLAST search. The predicted amino acid sequences of *MGG_08101* exhibited high levels of similarity to filamentous ascomycetes, showing high homology with EWZ90470 of *Fusarium oxysporum* f. sp. *lycopersici* (65.87%), XP_011394869 of *Neurospora crassa* (63.24%), XP_011319317 of *F. graminearum* (61.35%), and XP_024550716 of *Botrytis cinerea* (57.06%). Therefore, they formed a phylogenetic clade (Figure 1a). Phylogenetic analyses of *MGG_08101* homologues showed that they were conserved in eukaryotes and clustered into a group related to fungi, animals, and plants, with the exception of *Saccharomycetaceae* (Figure 1a).

ScNot3 and ScNot5 were compared to MGG_08101 at the protein level. The MGG_08101 locus encodes 662 amino acids and shows 32.4% and 29.0% overall identity to ScNot3 and ScNot5, respectively (Figure 1b). After comparing the conserved protein sequences of the N-terminal regions [8], MGG_08101 showed 62.1% and 69.0% similarity with ScNot3 and ScNot5, respectively. Based on these results, we designated the protein encoded by MGG_08101 as *MoNOT3* (*Magnaporthe oryzae NOT3*).

To investigate the interaction of *MoNOT3* with *M. oryzae*, 48 genes that predicted the first neighbours of *MoNOT3* were listed in a diverse database. *MoNOT3* interacts with many genes involved in transport, mycelium development, metabolism, signalling, transcription, protein localisation and modification, translation, DNA replication and chromatin-related processes, mRNA processing, exo- and endocytosis, and pathogenesis. The results showed that the *MoNOT3* genes were most closely related to *MGG_06958*, *MGG_01228*, *MGG_07928*, and *MGG_06044* (Figure 2a).

qRT-PCR was used to examine the expression pattern of *MoNOT3* from the mycelia to the infection stages. The *MoNOT3* gene was constitutively expressed at all stages (Figure 2b). This pattern of *MoNOT3* expression appears to correlate with its functional role in fungal development and pathogenicity.

### 2.2. Targeted Replacement of MoNOT3

Deletion mutants were generated via homologous recombination using a resistance marker cassette (Figure 3a). The protoplasts were transformed using genetic constructs. Two *Δmonot3* deletion mutants were identified based on Southern blot analysis. Genomic DNA samples were digested with *Eco*RI and probed with the 1.5 kb 3′-flanking regions (Figure 3a). The 3.9 kb band was detected in *Δmonot3-1* and *Δmonot3-2*, but not in WT KJ201 (Figure 3b). The *MoNOT3* transcript was detected in the RT-PCR and qRT-PCR analyses of the WT and complemented strains MoNOT3c-1 and MoNOT3c-2, but was not detected in *Δmonot3-1* and *Δmonot3-2* (Figure 3c).

### 2.3. MoNOT3 Is Required for Vegetative Growth and Conidiogenesis

To characterize the functional role of *MoNOT3*, the mycelial growths of the WT, mutant strain (*MoNOT3*), and the complement strain (MoNOT3c) were compared on CM, MM, MM-N, and MM-C. Mycelial growth rates were significantly reduced in the mutants *ΔΔmonot3-1* and *Δmonot3-2* compared to the WT on all four media. The complements (MoNOT3c-1 and MoNOT3c-2) were completely recovered, as shown in Figure 4.

The mutant showed a significant (>95%) reduction in conidiation compared to the WT on oatmeal agar medium (Figure 5a). Microscopic observations indicated that the WT strain formed dense conidiophores bearing conidia in a sympodial pattern, whereas the *Δmonot3-1* and *Δmonot3-2* mutants produced rare conidiophores, each consisting of a single conidium. These results indicate that *MoNOT3* is required for both conidiophore and conidial development. Both *Δmonot3-1* and *Δmonot3-2* mutants showed a significant reduction in the production of conidia compared with the WT. However, conidiation fully recovered to the WT levels after complementation with *MoNOT3* in the MoNOT3c-1 and MoNOT3c-2 strains (Figure 5b).

Microscopic observations revealed that the gene deletion mutants (*Δmonot3-1* and *Δmonot3-2*) produced conidia with abnormal shapes and septation defects (Figure 5c). While WT conidia were typically three-celled, the *Δmonot3-1* and *Δmonot3-2* strains produced more than 50% and 45% of the one-celled and two-celled conidia, respectively (Figure 5d). Furthermore, quantification of the abnormal shape defect revealed that the length and width of conidia in *Δmonot3-1* and *Δmonot3-2* strains were significantly smaller than those in the WT and complements (Figure 5e). These data indicate that *MoNOT3* is required for proper asexual reproduction in *M. oryzae*.

### 2.4. MoNOT3 Is Required for Conidial Germination and Appressorium Formation

Conidial germination and appressorium formation decreased in both *Δmonot3-1* and *Δmonot3-2* mutants (Figure 6a–c). Approximately 90% of WT conidia germinated at 8 h and 24 h following inoculation on a coverslip, whereas only 30% and less than 50% of *Δmonot3-1* and *Δmonot3-2* mutant conidia germinated after 8 h and 24 h of incubation, respectively. Conidial germination of the mutant did not fully recover even after culturing on a coverslip for a longer period (Figure 6a,b).

Over 95% of the germinated conidia formed appressoria at 8 h and 24 h in the WT and complemented strains, and the majority of germinated conidia in *Δmonot3-1* and *Δmonot3-2* failed to form appressoria (Figure 6a). After 24 h, the germ tubes grew longer and did not form appressoria in either the *Δmonot3-1* or *Δmonot3-2* mutants (Figure 6a,c). The complemented strains fully recovered germination and appressorium formation, similar to the WT strain.

### 2.5. MoNOT3 Is Required for Penetration and Virulence in M. oryzae

The results of the spray inoculation test showed that WT plants exhibited typical diamond-shaped lesions. In contrast, the *Δmonot3-1* and *Δmonot3-2* mutants exhibited a complete failure of disease development. Complements containing the *MoNOT3* gene were fully recovered (Figure 7a,b, left). The detached leaf test also produced results identical to those of spray inoculation (Figure 7a,b, middle). To gain a deeper understanding of this mechanism, we investigated invasive growth by injecting a conidial suspension into rice seedlings via infiltration. The mutants showed no blast symptoms (Figure 7a,b, right).

To assess the penetration ability of *MoNOT3* in plants, a suspension of conidia was applied to the onion epidermis (Figure 7c) and inoculated onto rice sheath cells (Figure 7d). The mutants showed no penetration or invasive hyphae in either non-host or host plants (Figure 7c,d). The complemented strains showed fully restored pathogenicity, invasive growth, and penetration, similar to the WT.

### 2.6. MoNOT3 Is Important for Stress Tolerance

In yeast, mutants lacking NOT are defective under thermal and cell wall stress conditions. To investigate the role of *MoNOT3* in various stress-resistance mechanisms, we measured the inhibition rates of mycelial growth compared to growth on CM. Nine days after inoculation, the mycelial growth of *Δmonot3-1* and *Δmonot3-2* mutants was significantly reduced compared to that of the WT on CM containing Congo red and SDS. The mutant was sensitive to cell wall-damaging agents in the presence of Congo red and SDS.

To examine the role of *MoNOT3* under thermal stress conditions, we analysed the mycelial growth at an optimal temperature of 25 °C and a maximum temperature of 32 °C. The WT grew normally at 32 °C, while both *Δmonot3-1* and *Δmonot3-2* mutants did not. The mycelial growth of *Δmonot3-1* and *Δmonot3-2* mutants exhibited a significant decrease compared to the WT at 32 °C. These results indicate that *Δmonot3* is involved in thermal stress resistance (Figure 8 and Table 1).

### 2.7. Analysis of MoNOT3 Gene Expression Patterns

To identify *MoNOT3* genes that potentially control conidiation, we analysed the expression patterns of conidia-related genes compared to the WT during conidiation. The expressions of *Con1*-related conidiophore stalk and *Con7*-related conidial morphology were downregulated during conidiation (Figure 9a). However, transcription factor genes related to conidia were not affected by *MoNOT3*. These results suggest that conidiation and conidial morphology are affected by *MoNOT3*. As mentioned earlier, we found that *MoNOT3* interacts with numerous genes involved in all stages.

To identify protein interactions involving *Δmonot3*, we quantified the transcripts of four genes associated with ubiquitin proteins. Polyubiquitin expression increased during mycelial growth (Figure 9b). These results suggest that *MoNOT3* regulates ubiquitination.

## 3. Discussion

CCR4-NOT complexes are essential proteins that regulate and coordinate diverse biological processes. While numerous new studies have been published on the functional analyses of CCR4-NOT complexes in yeast, humans, human pathogenic fungi, and some phytopathogenic fungi, the characterisation of CCR4-NOT complexes in phytopathogenic fungi is still required [2,6]. In this study, we characterised the crucial regulatory roles of *MoNOT3* in the developmental stages of *M. oryzae*. The absence of the *MoNOT3* gene led to multiple defects in mycelial growth, conidiation, conidial morphology, germination, appressorium formation, cell wall stress response, and virulence.

The CCR4-NOT complexes are well conserved in *M. oryzae* (Figure 1a). The *MoNOT3* gene is homologous to the yeast genes *ScNOT3* and *ScNOT5*, human gene *CNOT3*, *C. albicans* gene *CaNOT5*, and *F. graminearum* gene *FgNOT3* [12]. In *C. albicans*, *CaNOT5* is involved in spore attachment and pathogenicity [9,10]. In *F. graminearum*, *FgNOT3* is involved in mycelial growth, conidiation, sexual reproduction, and the production of secondary metabolites [12]. In the present study, we discovered that *MoNOT3* is involved in infection-related cell development, including conidiation, conidial germination, appressorium formation, invasive growth, spore germination, and appressorium formation. These results indicate that *MoNOT3* may be involved in all stages of fungal development.

Interestingly, we found that *MoNOT3* contributes to conidiation-related morphogenesis, including the development of conidiophore stalks, conidial morphology, and conidial production. Except for germination, these phenotypes were similar to those observed in other fungi [12]. The deletion of *FgNOT3* results in defective conidial morphology and germination [12]. However, the germination of *ΔFgnot3* fully recovered over time, whereas that of *Δmonot3* did not, even with additional time. These differences may indicate species divergence among filamentous fungi. Because delayed mycelial growth, germination, and abnormal septation are related to cell cycle progression, the observed phenotypes in the mutants suggest that *MoNOT3* is involved in regulating the cell cycle.

Previous studies have reported that mutants of *C. albicans* and *F. graminearum* were deficient in virulence. The deletion of *CaNOT5* does not induce a switch to a hyphal transition related to virulence [9]. In *F. graminearum*, deletion of *FoNOT3* resulted in an increase in secondary metabolites but did not lead to disease development [11]. The mutants were unable to form mats or aspersoria-like structures to penetrate the host cell wall. Unlike *F. graminearum*, in this study, we found that *MoNOT3* was associated with appressorium formation and invasive hyphae, demonstrating its crucial role in penetration. This is the first report indicating that *MoNOT3* is involved in the formation of the appressorium and invasive hyphae for penetration.

In yeast, double knockout mutants of *NOT1*, *NOT2*, and *NOT3/NOT5* are sensitive to heat stress [29]. In *F. graminearum*, *NOT* module mutants exhibit sensitivity to heat stress [12]. *CaNOT5* in *C. albicans* was sensitive to calcofluor white, whereas *FoNOT2* was sensitive to calcofluor white, Congo red, and oxidative stress. *Δmonot3* mutants were sensitive to heat stress, SDS, and Congo red. However, there was no difference in the other cell wall stress conditions. These differences may indicate species divergence among filamentous fungi. Other modules appear to play a more significant role in cell wall stress conditions in fungi than *Δmonot3*.

To understand these phenotypes, we identified the proteins that interact with *MoNOT3*. These results suggest that *MoNOT3* interacts with ubiquitin, which plays an important role in various cellular processes and host–pathogen interactions [30]. The levels of polyubiquitin transcripts increased in the mutant. Mono-ubiquitination may negatively affect polyubiquitin levels, and our transcript analyses demonstrate interactions among these proteins. In *M. oryzae*, *Ssb1*, *Ssz*, *Zuo1*, and *Mkk1*, which are associated with heat shock proteins, are involved in growth, pathogenicity, cell wall integrity, and conidiation [31]. The loss of *Ssb1* resulted in protein misfolding. When comparing the phenotype of *Δmonot3* with *ΔMoSsb1*, it is necessary to confirm the interactions with heat shock proteins. Monosodium triphosphate may affect the expression of heat shock proteins, which are crucial regulators.

## 4. Materials and Methods

### 4.1. Fungal Isolate and Culture Conditions

*M. oryzae* wild-type (WT) strain KJ201 and all the generated transformants in this study were cultivated on V8-Juice agar medium (8% Campbell’s V8-Juice, 1.5% agar, pH adjusted to 7 using 0.1 N NaOH) or oatmeal agar medium (50 g oatmeal per litre) at 25 °C under constant fluorescent light for 7 days to induce the production of conidia [18]. To extract genomic DNA and RNA, mycelia were isolated from fungal mycelia collected from 4-day-old cultures cultivated in liquid complete medium (CM, 0.6% yeast extract, 0.6% casamino acid, and 1% sucrose) for 5 days.

To assess mycelial growth, we used modified complete agar medium (mCM, 10 g of sucrose, 2 g of peptone, 1 g of yeast extract, 1 g of casamino acids, 6 g of NaNO_3_, 0.5 g of KCl, 0.5 g of MgSO_4_, 1.5 g of KH_2_PO_4_, and 15 g of agar per litre supplemented with 1 mL of trace elements (containing per litre: 22 g of ZnSO_4_·7 H_2_O, 11 g of H_3_BO_3_, 5 g of MnCl_2_·4 H_2_O, 5 g of FeSO_4_·7 H_2_O, 1.7 g of CoCl_2_·6 H_2_O, 1.6 g of CuSO_4_·5 H_2_O, 1.5 g of Na_2_MoO_4_·2 H_2_O and 50 g of Na_2_EDTA, at pH 6.5 adjusted by 1 M KOH)) and 1 mL of vitamin solution (containing per litre: 0.1 g of biotin, 0.1 g of pyridoxin, 0.1 g of thiamine, 0.1 g of riboflavin, 0.1 g of p-aminobenzoic acid, and 0.1 g of nicotinic acid), minimal agar medium (MM, mCM without peptone, yeast extract, and casamino acids, pH 6.5), carbon-starved medium based on MM (MM-C, mCM without peptone, yeast extract, casamino acids, and glucose, pH 6.5), and nitrogen-starved medium based on MM (MM-N, mCM without peptone, yeast extract, casamino acids, and NaNO_3_, pH 6.5), as previously described [32]. Genomic DNA was isolated from mycelia collected from 4-day-old cultures grown in liquid CM at 25 °C under dark for 7 days.

To investigate the role of *MoNOT3* in various stress conditions, including salt, cell wall, oxidative and thermal stresses, different stressors were introduced to the CM agar medium. Specifically, CaCl_2_ (0.2 M), NaCl (0.5 M), MnCl2 (0.02 M), sodium dodecyl sulphate (SDS) (0.01%), Congo red (200 ppm), calcofluor white (200 ppm), and H_2_O_2_ (2 mM) were added to assess their impacts on fungal mycelial growth. Thermal stress response assays were performed at 25 °C and 32 °C on CM. The inoculum was prepared with 5 mm agar plugs from 7-day-old strains on CM, and the plugs were then transferred upside down onto each medium. Mycelial growth was measured after incubation at 25 °C under dark for 9 days, and the percentage of inhibition was calculated by comparing the mycelial growth on the CM.

### 4.2. Computational and Phylogenetic Analyses

The MoNOT3 protein sequence (MGG_08101) and sequences of *NOT3* various fungi were obtained from the Comparative Fungal Genomics Platform “http://www.cfgp.snu.ac.kr (accessed on 20 March 2021)” [33]. Homology searches for DNA and protein sequences were performed using BLAST at the National Center for Biotechnology Information (NCBI). Sequence analysis and domain architecture of NOT3/5 were conducted using InterPro Scan, which is available at the European Bioinformatics Institute “http://www.ebi.ac.uk/interpro/ (accessed on 31 August 2021)”. The amino acid sequence of *NOT3* was aligned using ClustalW “http://align.genome.jp/ (access on 30 April 2021)”. Phylogenetic trees were constructed using the maximum likelihood method in the MEGA program [34]. All sequence alignments were tested using the bootstrap method with 1000 repetitions.

### 4.3. Vector Construction and Fungal Transformation

The construction of the *MoNOT3* gene knockout mutant was performed using double-joint polymerase chain reaction (PCR) [35]. Fragments corresponding to 1.5 kb upstream and downstream of *MoNOT3* were amplified using primers MoNOT3_5UF/5UR and MoNOT3_DF/DR, respectively. A 1.4-kb hph cassette was amplified with primers HygB_F and HygB_R (Table 2) from pBCATPH, which contains the hygromycin phosphotransferase gene (*hph*) under the control of the *Aspergillus nidulans TrpC* promoter [36]. The *MoNOT3* gene disruption construct was amplified with primers nested in MoNOT3_5UF/3DR using the fused products as a template and was directly transformed into the protoplasts of strain WT.

Protocols for protoplast preparation and fungal transformation were adapted from a previous study [37]. Hygromycin-resistant transformants were screened by PCR using primers MoNOT3_ORF_F/R (Table 2).

For complementation, a 4-kb fragment containing the native promoter and ORF of *MoNOT3* was amplified with primers MoNOT3_5UF and 3DR (Table 2) from the genomic DNA of the WT. This fragment was co-transformed into protoplasts from the *MoNOT3* gene deletion mutant using pll99, which contains a gene that confers resistance to geneticin. Transformants were selected based on their ability to grow in the presence of geneticin (800 μg/mL).

### 4.4. Nucleic Acid Manipulation and Southern Blot Analysis

Genomic DNA was isolated by using the NucleoSpin^®^ Plant II Kit (MachereyNagel, Düren, Germany). The DNA concentration was estimated using a NanoDrop ND-1000 spectrophotometer (NanoDrop Technologies, Inc., Wilmington, NC, USA). Putative *MoNOT3* gene deletion mutants were confirmed by Southern blot analysis. For Southern blotting, genomic DNA (approximately 5 μg) was digested with *Eco*RI and fractionated on a 0.7% agarose gel at 40 V for 4 h in 1% Tris-acetate-EDTA buffer. The fractionated DNA was transferred onto Hydond N+ nylon membranes (Amersham International, Little Chalfont, England, UK) using 10× SSC (1× SSC comprised 0.15 M NaCl plus 0.015 M sodium citrate). The probe, designed from the 3′-flanking region, was labelled with alkaline phosphate using AlkPhos Direct Labelling Reagents (Amersham Biosciences, Amhersheim, UK). Hybridisation, washing, and detection were performed according to the instruction manual of the Alkphos Direct Labelling and Detection System using CDP Star (Amersham Pharmacia, Piscataway, NJ, USA).

Briefly, hybridisation was performed at 55 °C in Gene Images AlkPhos Direct hybridisation buffer, including 0.5 M NaCl and 4% blocking reagent. After hybridisation, the membrane was washed twice in primary wash buffer at 55 °C for 10 min and then washed twice in secondary wash buffer at room temperature for 5 min. Target DNA was detected using CDP star reagent (Amersham International plc, Buckinghamshire, UK), according to the manufacturer’s instructions.

### 4.5. Conidiation, Conidial Morphology, Conidial Germination, and Appressorium Formation

To measure conidiation and conidial morphology, conidia were collected from 7-day sporulating cultures on V8-Juice agar media by gently rubbing the mycelia with sterilised distilled water, followed by filtration through Miracloth (Calbiochem, San Diego, CA, USA). The number of conidial suspensions was counted using a haemocytometer (BRAND counting chamber, BLAUBRAND, Brand GMBH, Frankfurt, Germany). Conidial size was measured using a microscope, and the conidia were stained with calcofluor white and KOH to observe conidial morphology [38].

Conidial germination and appressorium formation were measured using hydrophobic coverslips [25]. A total of 50 μL of a conidial suspension (5 × 10^4^ conidia/mL) was placed on a coverslip and incubated in the moist box at 25 °C for 8 h and 24 h. The percentage of conidia with germ tubes and appressoria was determined by microscopic examination of at least 100 conidia per replicate. All experiments were performed in triplicate.

### 4.6. Plant Infection Assays

For the pathogenicity assay, conidia were harvested from 7-day-old cultures on V8 agar plates, and 10 mL of a conidial suspension (1 × 10^5^ conidia/mL) containing 250 ppm Tween 20 was sprayed onto 4-week-old susceptible rice seedlings (*Oryza sativa* cv. Nakdongbyeo) at the three-to-four-leaf stage. For the detached leaf assay, 50 μL of conidial suspension (2 × 10^4^ conidia/mL) was dropped on three points per leaf of 4-week-old rice plans. For the infiltration infection assay, 100 μL of conidial suspension (2 × 10^4^ conidia/mL) was injected into three points per leaf of 4-week-old rice plants. All inoculated plants were kept in a dew chamber at 25 °C for 24 h in the dark and then moved to a growth chamber with a photoperiod of 16 h under fluorescent lights. Disease severity was measured 5 days after inoculation.

Plant penetration assays were performed using onion epidermis and rice sheaths, as previously described [39]. Briefly, 4-week-old rice sheaths of cv. Nakdongbyeo and conidia (1 × 10^4^ conidia/mL) from 7-day-old cultures on V8-Juice agar media were used for onion epidermis and rice sheaths. The inoculated plants were maintained in a moist box at 25 °C for 36 h in the dark. An infection assay using onion epidermis was performed using fresh onion epidermis. All experiments were performed in triplicate.

### 4.7. RNA Isolation, Quantitative Real-Time (qRT)-PCR, and Gene Expression Analysis

Total RNA was prepared using the Easy-Spin Total RNA Extraction Kit (iNtRON Biotechnology, Sungnam, Korea), following the manufacturer’s instructions. Total RNA was extracted from the fungus at several developmental stages under stress conditions and used as a template for reverse transcription (RT)-PCR. Total RNA was extracted from mycelia in liquid mCM for expression analysis of the *COS1*, *Com1*, *Con7*, and *MoHox2* genes.

To obtain samples during conidiogenesis, WT and gene-deletion mutants were inoculated into liquid mCM and incubated at 25 °C on a 120 rpm orbital shaker for 3 days. The resulting mycelia were fragmented using a spatula and filtered through a single-layer cheesecloth. Mycelia were collected using one layer of Miracloth (Calbiochem, San Diego, California, USA) in 2 mL of sterilised distilled water. Then, 400 μL of the mycelial suspension was spread onto individual 0.45 μm pore cellulose nitrate membrane filters (Whatman, Maidstone, England) placed on V8-Juice agar plates using a loop. The plates were dried and incubated at 25 °C with constant light. After 12 h of inoculation, whole mycelia on cellulose nitrate membrane filters were collected using a disposable scraper (iNtRON Biotechnology, Seoul, Korea) [27].

qRT-PCR reactions were performed using a 96-well reaction plate and a CFX96 Touch™ Real-Time PCR Detection System (Bio-Rad Laboratories, Hercules, CA, USA). Each well contained 5 μL of SYBR Green I Master mix (Bio-Rad Laboratories, Hercules, CA, USA), 2 μL of cDNA (12.5 ng) and 15 pmol of each primer. The thermal cycling conditions were 10 min at 94 °C, followed by 40 cycles of 15 s at 94 °C, and 1 min at 60 °C. All amplification curves were analysed with a normalised reporter threshold of 17 to obtain the threshold cycle (Ct) values.

The primer sequences used for each gene are listed in Table 2. To compare the relative abundance of the target gene transcripts, the average Ct value was normalised to that of *β-tubulin* (MGG00604) for each of the treated samples as 2^−Ct^, where Ct = (Ct, _target gene_ − Ct, _β-tubulin_). Fold changes during fungal development and infectious growth in liquid CM were calculated as 2^−Ct^, where Ct = (Ct, _target gene_ − Ct, _β-tubulin_)_test condition_ − (Ct, _WT_ − Ct, _β-tubulin_)_CM_ [27]. qRT-PCR was performed using three independent tissue pools with two sets of experimental replicates.

## 5. Conclusions

In our previous study, we analysed the function of the *MoNOT3* (*MGG_08101*) genes, which were up-regulated during appressorium formation in *M. oryzae*. For functional analysis, gene deletion mutants were generated and phenotype analysis was conducted. As a result, the mycelial growth was delayed compared to WT. In particular, it was found to be involved in various functions related to conidiation, including conidial morphology and conidial germination. Furthermore, it appears to have played a crucial role in the formation of appressoria and invasive growth, which are essential for invading the host plant. This suggests that *MoNOT3* may be specifically involved in the process of disease development. Since this protein was found to be involved in heat shock in *F. graminearum*, it was also found to be involved in various stress conditions in *M. oryzae*. Through transcriptional analysis in WT and *Δmonot3* mutant strains, the results indicate that the *MoNOT3* gene regulates COS1 and COM1, which are involved in conidiation. Taken together, *MoNOT3* is presumed to be a gene involved in various aspects of the disease development process of *M. oryzae*, as well as necessary for temperature regulation and ionic stress.

## Figures and Tables

**Figure 1 ijms-25-03290-f001:**
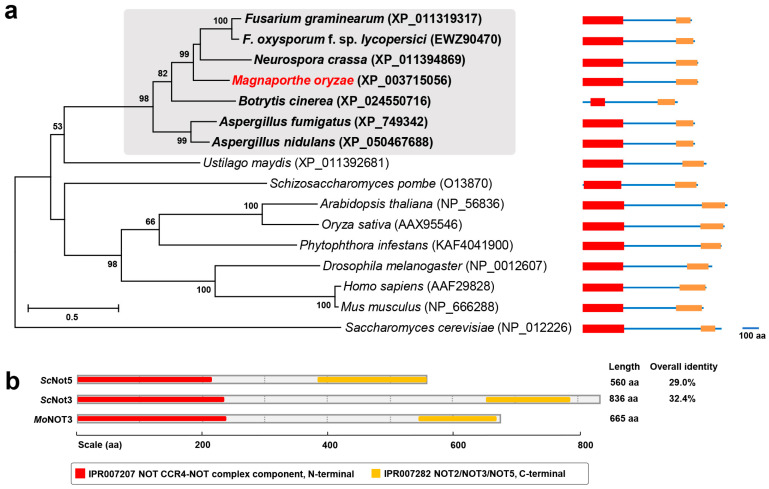
Phylogenetic trees. (**a**) Phylogenetic analysis of *NOT3* from *Magnaporthe oryzae* with eukaryotes. A maximum likelihood tree was constructed based on the amino acid sequence using MEGA X. All sequence alignments were tested with a bootstrap method using 1000 repetitions. The domain architecture of *NOT3* was determined by InterProScan. The scale bar shows the number of amino acid differences per site. Subclades containing *MoNOT3* are shaded, in which *MoNOT3* and characterised genes from other fungi are shown in bold in red or black, respectively. The text with numbers in parentheses represents the National Center for Biotechnology Information (NCBI) accession numbers of the protein sequences. (**b**) Schematic diagram of the conserved region s of Not3 subunit homologs between *S. cerevisiae* and *M. oryzae*. The percentage of identity between proteins was calculated using the MegAlign module of the DNASTAR program package (MegAlign 5.00, DNASTAR Inc., Madison, WI, USA).

**Figure 2 ijms-25-03290-f002:**
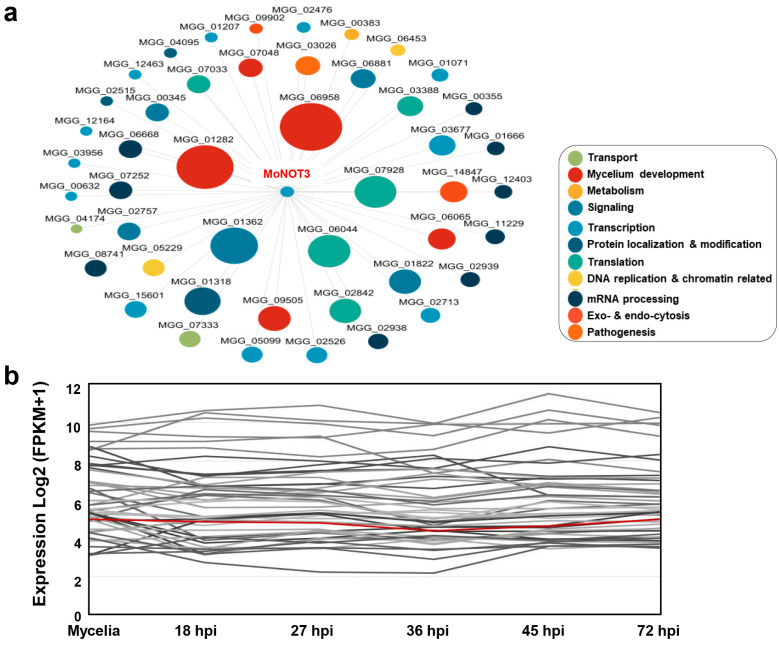
Putative protein–protein interaction (PPI) network and the expression patterns of *MoNOT3*. (**a**) The PPI network of *MoNOT3*. Only the first neighbours of *MoNOT3* are shown. The circle size of the nodes represents the number of edges (interactions) connected to each node. The putative function of each node represented by the GO term is colour coded. (**b**) The expression patterns of *MoNOT3* and its PPI neighbours. The expression patterns from mycelia to the infection stages of *MoNOT3* (red line) and its 47 neighbours are shown.

**Figure 3 ijms-25-03290-f003:**
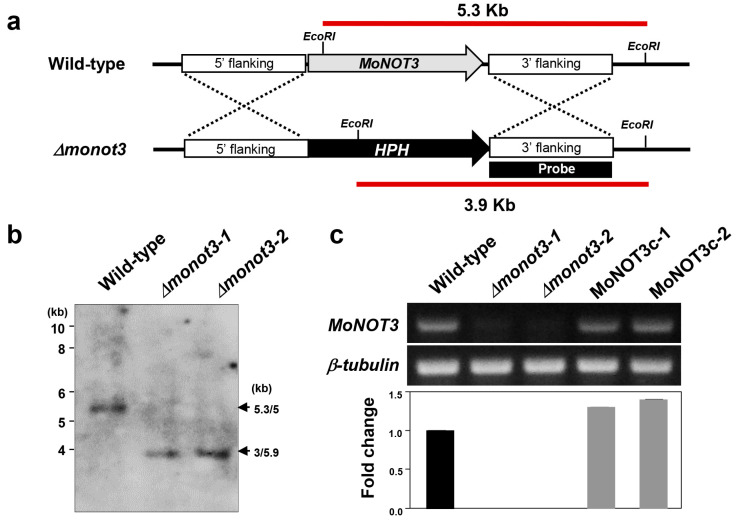
Generation of gene deletion mutants for the *MoNOT3* gene. (**a**) The *Δmonot3* knockout fragment was constructed by PCR amplification. The 1.5 kb upstream and downstream flanking sequences were amplified with primers *MoNOT3*_UF/UR and *MoNOT3*_DF/DR, respectively, and ligated with the hph gene cassette. (**b**) Southern blot analysis of *Eco*RI-digested genomic DNA of the wild-type (WT) and the *Δmonot3* mutants, *Δmonot3-1*, and *Δmonot3-2*. The blot was hybridized with the probe shown in (**a**). (**c**) Expression analysis of *MoNOT3* in the WT strain, *Δmonot3* mutants, and the complemented transformants, MoNOT3c-1 and MoNOT3c-2, using *β-tubulin* as control. Gene disruption resulted in the loss of the transcript harbouring the MoNOT3 (top). The expression of the *β-tubulin* gene was constrictively expressed in all the strains (middle). Quantitative real-time PCR also confirmed the loss of the MoNOT3 transcript in the *Δmonot3* mutants, but not wild-type and complements, MoNOT3-1, and MoNOT3-2 (bottom).

**Figure 4 ijms-25-03290-f004:**
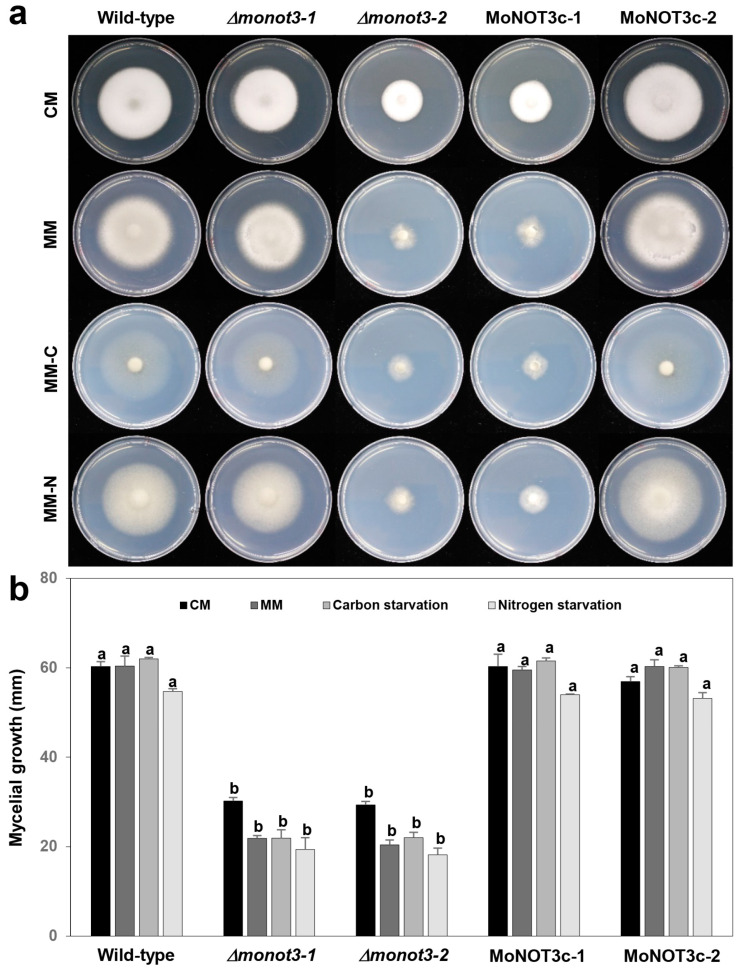
Hyphal growth of *M. oryzae* WT, *Δmonot3*, and MoNOT3c on various media. (**a**) Hyphal growth of *M. oryzae* was measured on complete medium (CM), minimal medium (MM), carbon-starved medium based on MM (MM-C), and nitrogen-starved medium based on MM (MM-C). (**b**) Mycelial growth (mm) was measured at 9 days post-inoculation on the different types of media. Significance was determined by Tukey’s test (*p* = 0.01). The same letters in a graph showed no significant difference.

**Figure 5 ijms-25-03290-f005:**
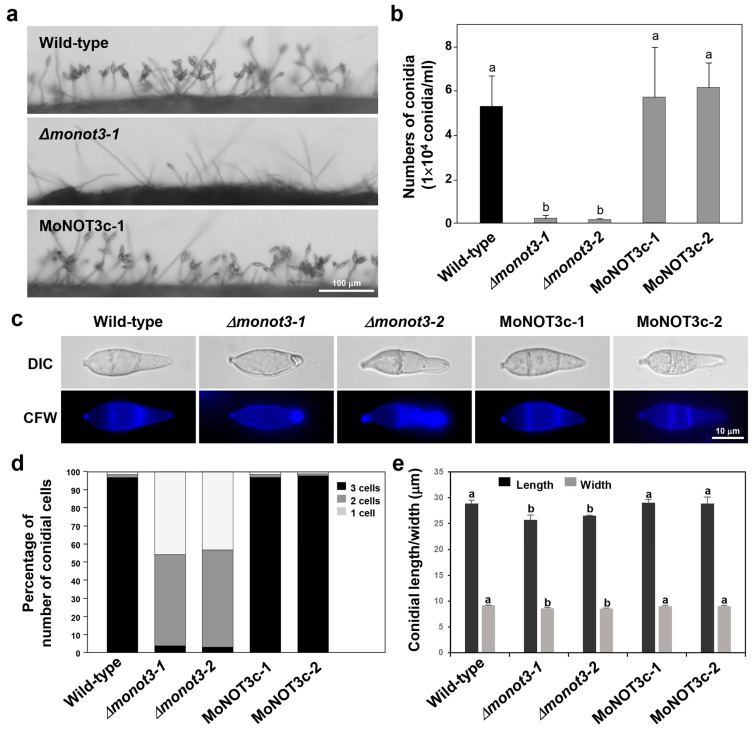
Effect of *MoNOT3* disruption on conidiation, production of conidia, and conidial morphology of *M. oryzae*. (**a**) Development of conidiophores and conidia was visualised in *M. oryzae* WT, *Δmonot3*, and MoNOT3c strains grown on oatmeal agar media. Conidiogenesis on conidiophores was observed under a microscope. Scale bar, 100 μm. (**b**) Production of conidia numbers in *M. oryzae* WT, *Δmonot3*, and MoNOT3c strains. Conidia of the WT, deletion mutants, and complemented strains were collected from V8 agar after incubation for 7 days. (**c**) Conidial shape and septum formation were analysed in *M. oryzae* WT, *Δmonot3*, and MoNOT3c strains. Conidia were stained by calcofluor white with KOH. (**d**) Distribution of conidia cell numbers in the control and *Δmonot3* mutant strains. Conidia (>100) of each strain were counted by microscopic observation. (**e**) Conidial size of the WT, *Δmonot3*, and MoNOT3c strains. Values are the means ± SD from >100 conidia of each strain, which were measured using the Axiovision image analyser. Tukey’s test was used to determine significance at the 95% probability level. The same letters show no significant differences.

**Figure 6 ijms-25-03290-f006:**
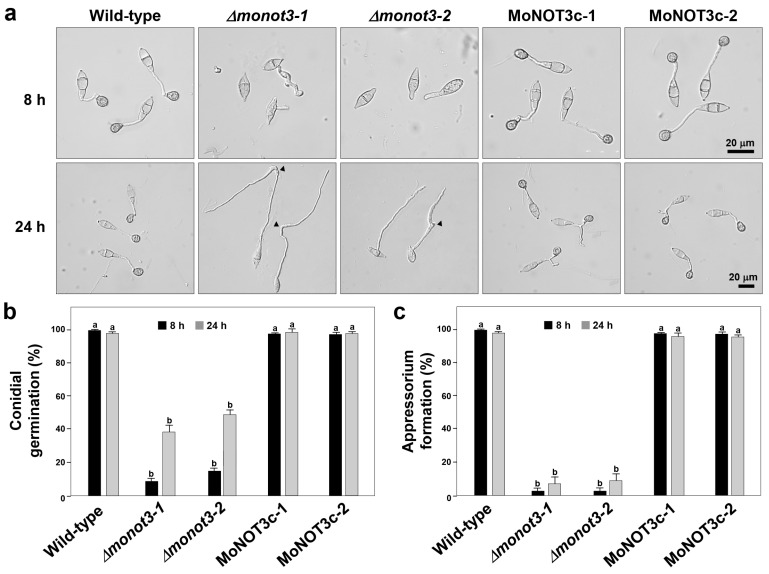
Effect of *MoNOT3* deletion on conidial germination and appressorium formation. (**a**) Microscopic observation of appressorium development on germ tubes. Appressorium formation was examined at 8 and 24 h after incubation of the conidial suspension on hydrophobic cover slips. Black arrowheads in *Δmonot3* indicate the occurrence of swellings and hooks in a germ tube. Percentage of conidial germination (**b**) and appressorium formation (**c**) on a hydrophobic coverslip was measured at 8 and 24 h after incubation of the conidial suspension under a light microscope using conidia harvested from 6-day-old V8-Juice agar medium. Tukey’s test was used to determine significance at the 95% probability level. The same letters show no significant differences in the graph.

**Figure 7 ijms-25-03290-f007:**
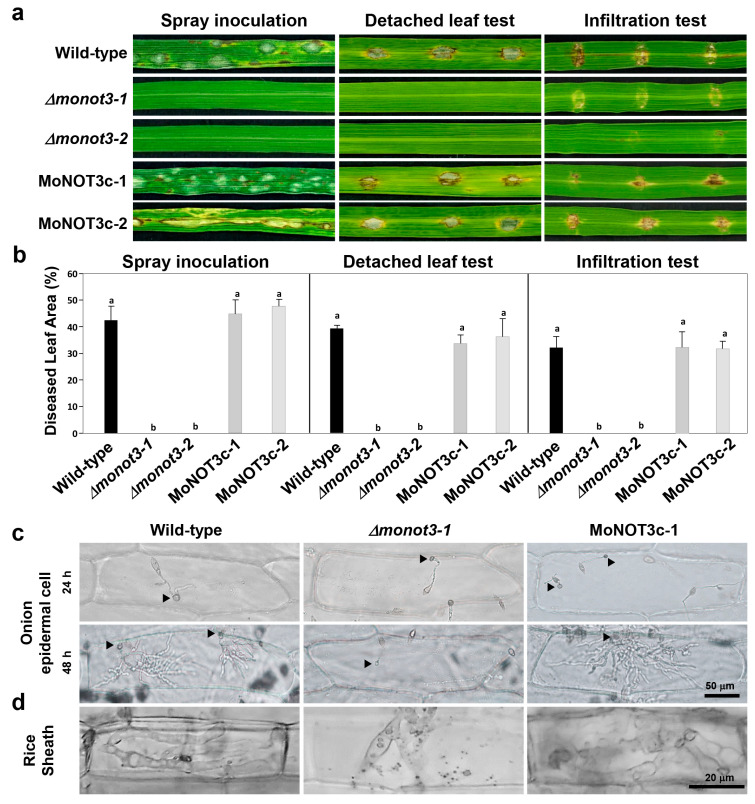
Pathogenicity of *Δmonot3*. (**a**) Pathogenicity assay results. Spray inoculation (**left**). Conidial suspensions (1 × 10^5^/mL) were sprayed onto 4-week-old rice seedlings, and lesions were observed at 5 days post-inoculation. Detached leaf inoculation (**middle**). Conidial suspensions (2 × 10^4^/mL) were dropped onto 4-week-old rice seedlings. Infiltration inoculation (**right**). Conidial suspensions (2 × 10^4^/mL) were injected onto 4-week-old rice seedlings. (**b**) Diseased leaf area (DLA) percentage was measured for each tested pathogenicity result using ImageJ 1.48v software. The same letters show no significant differences in the graph. (**c**) Onion epidermal cell infection 24 and 48 h after inoculation. (**d**) Rice sheath infection 48 h after inoculation. The growth of invasive hyphae was observed under a microscope at 48 h post-inoculation. Arrows indicate appressoria.

**Figure 8 ijms-25-03290-f008:**
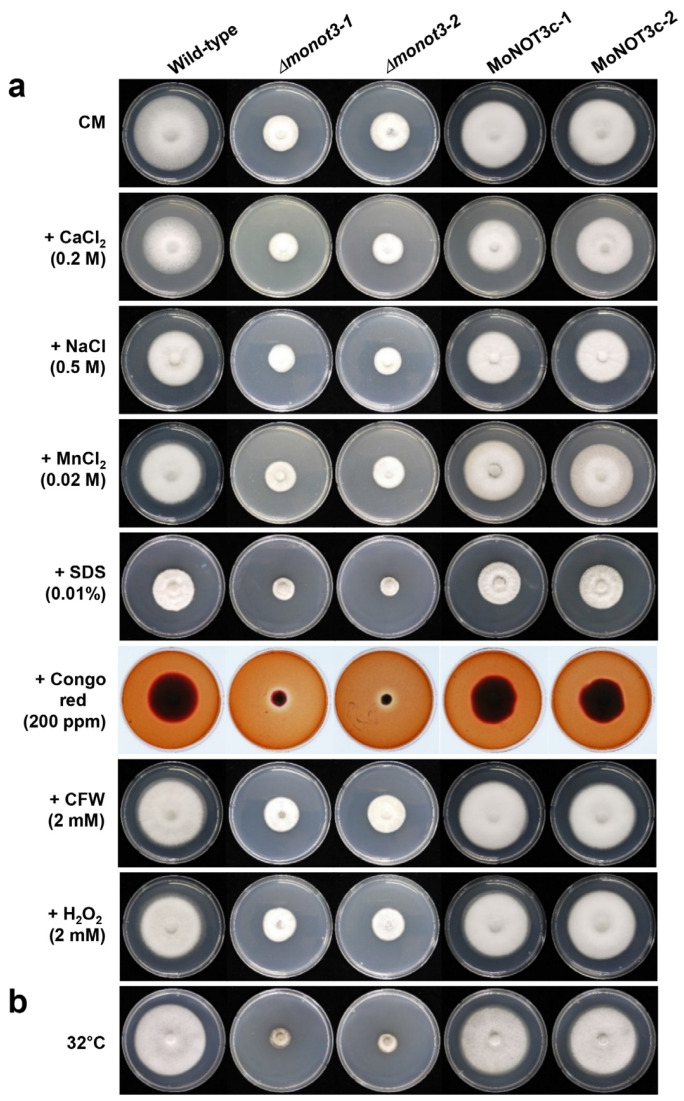
Hyphal growth of *M. oryzae* strains in this study under various stress conditions. (**a**) The WT, *ΔΔmonot3* (*Δmonot3-1* and *Δmonot3-2*), and complemented (MoNOT3c-1 and MoNOT3c-2) strains were grown for 9 days on CM and CM containing CaCl_2_ (0.2 M), NaCl (0.5 M), MnCl_2_ (0.02 M), sodium dodecyl sulphate (SDS) (0.01%), Congo red (200 ppm), calcofluor white (CFW) (200 ppm), and H_2_O_2_ (2 mM) at a temperature of 25 °C. (**b**) Hyphal growth of *M. oryzae* strains was observed to be higher at a relatively high temperature of 32 °C compared to 25 °C for 9 days on CM. The 5 mm-diameter mycelial blocks were transferred onto the media and incubated.

**Figure 9 ijms-25-03290-f009:**
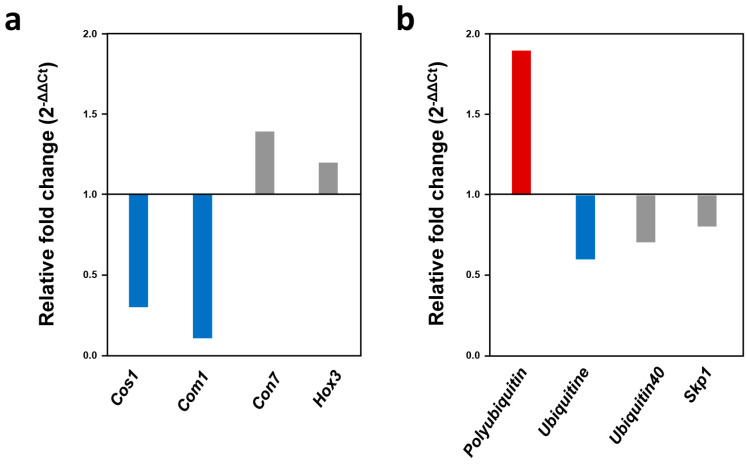
Expression of the WT and *Δmonot3* in *M. oryzae*. (**a**) Relative expression of conidiation-related genes during conidiation in the *Δmonot3*. (**b**) Relative expression of ubiquitin-related genes during mycelial growth in the *Δmonot3*. Transcript expression was quantified by qRT-PCR analysis, and values were normalized to the expression of *β*-*tubulin*. Relative expression levels, presented as fold changes (2^−ΔΔCt^), were calculated by comparison with the WT strain. Up-regulated genes (more than 1.5-fold) are indicated by red bars, and down-regulated genes (less than 0.5-fold) are noted by blue bars. The genes that did not show differential expression are marked in grey.

**Table 1 ijms-25-03290-t001:** Hyphal growth of *M. oryzae* strains under diverse stress conditions.

Strain	Hyphal Growth ^A^ (mm)								
	CM	CaCl_2_ (0.2 M)	NaCl (0.5 M)	MnCl_2_ (0.02 M)	SDS (0.01%)	Congo Red (200 ppm)	CFW (2 mM)	H_2_O_2_ (2 mM)	32 °C
Wild-type	69.3 ± 0.4 ^aB^	50.9 ± 1.9 ^a^	44.4 ± 1.7 ^a^	53.0 ± 1.7 ^a^	30.6 ± 0.3 ^a^	45.4 ± 0.8 ^a^	68.3 ± 0.2 ^a^	58.9 ± 1.5 ^a^	65.3 ± 0.2 ^a^
*Δmonot3* *-1*	34.3 ± 0.3 ^b^	26.0 ± 0.4 ^b^	22.2 ± 0.5 ^b^	27.7 ± 0.8 ^b^	11.3 ± 0.6 ^b^	12.2 ± 0.2 ^b^	33.2 ± 0.4 ^b^	29.8 ± 1.0 ^b^	17.4 ± 0.4 ^b^
*Δmonot3* *-2*	32.7 ± 0.6 ^b^	25.9 ± 0.4 ^b^	23.8 ± 0.8 ^b^	28.1 ± 0.2 ^b^	14.0 ± 0.5 ^b^	11.4 ± 0.6 ^b^	33.7 ± 0.5 ^b^	30.1 ± 0.1 ^b^	17.3 ± 0.3 ^b^
MoNOT3 c-1	68.3 ± 0.6 ^a^	49.9 ± 1.3 ^a^	44.2 ± 1.3 ^a^	52.8 ± 1.1 ^a^	32.1 ± 0.6 ^a^	42.3 ± 1.9 ^a^	67.2 ± 0.4 ^a^	59.1 ± 0.2 ^a^	65.2 ± 0.3 ^a^
MoNOT3 c-2	67.7 ± 0.2 ^a^	46.7 ± 0.8 ^a^	43.0 ± 1.4 ^a^	50.8 ± 1.1 ^a^	30.0 ± 1.0 ^a^	45.7 ± 1.2 ^a^	68.0 ± 0.5 ^a^	58.9 ± 1.4 ^a^	65.3 ± 0.3 ^a^

^A^ A 5 mm diameter mycelial plug was inoculated to complete media (CM) or CM supplemented with the indicated chemicals and then incubated at 25 °C, except for the experiment on temperature effects at 32 °C. Radial growth was measured at 9 days; ^B^ Tukey’s test was used to determine significance at the 95% probability level. The same letters in a column showed no significant difference.

**Table 2 ijms-25-03290-t002:** Primers used in this study.

Target	Name	Sequence (5′-3′)
Knockout	MoNOT3_5UF	ATATACAGGAGGCGGGGTCAGAGT
construct	MoNOT3_5UR	CCTCCACTAGCTCCAGCCAAGCCGGCTAGTTGTTGTTTCGGATGTCT
	MoNOT3_3DF	GTTGGTGTCGATGTCAGCTCCGGAGGAGTGAATACGGCCAATAGC
	MoNOT3_3DR	GCACAGAGCCTAACATCAAACCCC
	MoNOT3_5UF_nested	GACACCATGAAACCACGCACTCTA
	MoNOT3_3DR_nested	GGACCAACAAGCTCCTCTCA
hph gene	HygB_F	TCAGCTTCGATGTAGGAGGG
	HygB_R	TTCTACACAGCCATCGGTCC
Screening of	MoNOT3_ORF_F	TACATACGCCACTACCAACCATTA
transformant	MoNOT3_ORF_R	GATGCTGGAAGGTGGTAGATAGAT
qRT-PCR	*Mo β-tubulin*_F	ACAACTTCGTCTTCGGTCAG
	*Mo β-tubulin*_R	GTGATCTGGAAACCCTGGAG
	*MoNOT3*_qRT_F	CAAATGGAGAGGTTCAAAGCG
	*MoNOT3*_qRT_R	TGTCGATCATGTTCCCAAGG
	*COS1*_qRT_F	TGCACCACGATCCCAGAGA
	*COS1*_qRT_R	GCGATGTTGTGCCGTTGTTCC
	*COM1*_qRT_F	GCCAGAGGTCCGCTATCAAA
	*COM1*_qRT_R	CGGGATCTCGTCACTGGATT
	*CON7*_qRT_F	TAAGGAGATCCGCAAAGAGT
	*CON7*_qRT_R	TAGCGTTGTAGTCGGGGAGT
	*HOX2*_qRT_F	TGGGGTTCTGCAGCCATGTT
	*HOX2*_qRT_R	GTCCCGTGGTGTTACGTTCTGG
	*Polyubiqutin*_qRT-F	CAACGCCTTATTTTCGCTGG
	*Polyubiqutin*_qRT-R	TCTTGCCCGTCAAAGTCTTG
	*Ubiqutin*_qRT-F	GCAAGTTCAACTGCGACAAG
	*Ubiqutin*_qRT-R	TCCACACTTTCTCTTCCTGC
	*Ubiqutin40*_qRT-F	AGTTGGCTGTGCTCAAGTAC
	*Ubiqutin40*_qRT-R	GTAGGTCAGGTGGCAACG
	*Skp1*-qRT-F	TGGGATCAGAAGTTCATGCAG
	*Skp1*-qRT-R	ATATCAAGGTAGTTGCTCGCC

## Data Availability

Data are contained within the article.

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
