# Peer review of "MoNOT3 Subunit Has Important Roles in Infection-Related Development and Stress Responses in Magnaporthe oryzae"

_ijms, 2024, doi:10.3390/ijms25063290_

Round 1

Reviewer 1 Report

Comments and Suggestions for Authors

MoNOT3/5 subunit is important roles in infection-related development and stress responses in Magnaporthe oryzae

Specific remarks:

Materials and methods:

Page 3, Line 101 - Cultivation conditions are not written in detail.

Page 3, Line 105 - „under constant light“. Why under constant light?

Page 3, Line 110 - Reference 29. I cannot find all the mentioned medium in the specified work (29). I ask the authors to write in detail the composition of each medium they have listed.

Page 3, Line 111 - „4-day-old cultures grown in liquid CM.“ Write the cultivation conditions.

Page 3, Line 114 - „their effects on fungal mycelial growth.“ Why is their influence on the growth of mycelia investigated? Explain to be understandable.

Page 3, Line 117 - „Mycelial growth was measured after incubation for 9 days...“ What temperature? Under what conditions?

Page 5, Lines 170-177 - Provide a reference for the method.

Page 5, Line 195 - Reference 35. The reference is not adequate. I can't find the description of the method.

Results:

Page 10, Line 333 - Too many symbols in the subtitle.

Page 13, Lines 393, 394 - Figure 7 “…and H2O2 (2 mM) at 393 a temperature of 32°C.” Was the experiment with H2O2 done at 32 °C? The title of the Figure 7 should be corrected to make it understandable. Separate the H2O2 experiment and the experiment performed at 32 °C in the title of the Figure 7.

Page 13, Lines 394, 395 - Figure 7. Considering that one experiment from Figure 7 was done at 32 °C, it cannot be written that all of them were done at 25 °C. Please correct.

Page 14, Line 398 - Given that one experiment shown in Table 2 was done at 32 °C, it cannot be written that all of them were done at 25 °C. Please add an explanation for that experiment. The explanation of the Table must be detailed.

Page 14, Line 399, 400 - Table 2 “The same letters in a column showed no significant difference.” All letters showing differences in statistical significance should be superscripted.

 References

Nine references are considered to be quite out of date. It would not be bad if the authors replaced them with some more recent references.

General remarks:

The paper is interesting and a lot of work and time has gone into it. Therefore, I would like to commend the authors.

General opinion:

With minor revisions, I suggest that the paper be considered for publication.

Best Regards,

Reviewer

Author Response

Dear reviewer,

We are grateful for the chance to improve our manuscript with revision. We have fully revised the manuscript. Our responses are highlighted in blue.

We hope this revised manuscript is acceptable for publication in International Journal of Molecular Science. If you need any additional information or recommend any additional changes, please let me know. Once again, we appreciate your effort to improve the quality of our manuscript.

Sincerely,

Sook-Young Park

Reviewer 1

Materials and methods:

Page 3, Line 101 - Cultivation conditions are not written in detail.

-- In the case of Magnaporthe oryzae, when cultivated under constant light conditions at 23-27°C, spores are produced within 7-14 days, making this the most common method. All cultivations were conducted on V8-Juice agar medium and oatmeal agar medium. In this study, no special cultivation conditions were applied.

Page 3, Line 105 - „under constant light“. Why under constant light?

-- We added the information with reference.

Page 3, Line 110 - Reference 29. I cannot find all the mentioned medium in the specified work (29). I ask the authors to write in detail the composition of each medium they have listed.

-- We added the information.

Page 3, Line 111 - „4-day-old cultures grown in liquid CM.“ Write the cultivation conditions.

-- We added the information „ at 25°C under dark for 7 days“.

Page 3, Line 114 - „their effects on fungal mycelial growth.“ Why is their influence on the growth of mycelia investigated? Explain to be understandable.

-- We added the information.

Page 3, Line 117 - „Mycelial growth was measured after incubation for 9 days...“ What temperature? Under what conditions?

-- We added the information „ at 25°C under dark“.

Page 5, Lines 170-177 - Provide a reference for the method.

-- We added the references.

Page 5, Line 195 - Reference 35. The reference is not adequate. I can't find the description of the method.

 -- We replaced and added the updated reference.

Results:

Page 10, Line 333 - Too many symbols in the subtitle.

 -- We removed the typo.

Page 13, Lines 393, 394 - Figure 7 “…and H2O2 (2 mM) at 393 a temperature of 32°C.” Was the experiment with H2O2 done at 32 °C? The title of the Figure 7 should be corrected to make it understandable. Separate the H2O2 experiment and the experiment performed at 32 °C in the title of the Figure 7.

 -- We revised.

Page 13, Lines 394, 395 - Figure 7. Considering that one experiment from Figure 7 was done at 32 °C, it cannot be written that all of them were done at 25 °C. Please correct.

 -- We revised.

Page 14, Line 398 - Given that one experiment shown in Table 2 was done at 32 °C, it cannot be written that all of them were done at 25 °C. Please add an explanation for that experiment. The explanation of the Table must be detailed.

 -- We revised.

Page 14, Line 399, 400 - Table 2 “The same letters in a column showed no significant difference.” All letters showing differences in statistical significance should be superscripted.

 -- We revised.

Reviewer 2 Report

Comments and Suggestions for Authors

Magnaporthe oryzae is a phytopathogenic fungus that causes an extremely dangerous rice disease - rice blast, which annually causes great damage in all rice-growing regions. Yield losses can be 15-40%. Understanding the genetic mechanisms responsible for the ability of a fungus to infect a host plant is the basis for the development of new strategies for combating plant diseases, so the work of respected authors is of undoubted scientific and practical interest. This is a high-level, diverse study. The results obtained by the respected authors indicate that the studied gene MoNOT3/5 can be involved in all stages of fungal development. For the first time, the authors obtained data indicating the participation of MoNOT3/5 in the formation of appressoria and invasive hyphae of the fungus for penetration into the plant, which will undoubtedly make a significant contribution to the understanding of the mechanisms of pathogenesis of this harmful fungus.
The article is very well written, beautifully illustrated and will be of interest to a wide range of scientists. The “materials and methods” section is also very well written – clear, detailed and informative. The results obtained by the authors are beyond doubt.

I found only one small blot - line 333 - an extra bracket.

I believe that the article should be accepted for publication after minor revision.

Author Response

Dear reviewer,

We are grateful for the chance to improve our manuscript with revision. We have fully revised the manuscript. Our responses are highlighted in blue.

We hope this revised manuscript is acceptable for publication in International Journal of Molecular Science. If you need any additional information or recommend any additional changes, please let me know. Once again, we appreciate your effort to improve the quality of our manuscript.

Sincerely,

Sook-Young Park

Reviewer 2

I found only one small blot - line 333 - an extra bracket.

-- We removed an extra bracket at line 333.

Reviewer 3 Report

Comments and Suggestions for Authors

The article  MoNOT3/5 subunit is important roles in infection-related development and stress responses in Magnaporthe oryzae “ authored by Kim et al., chraterized the role of moNOT3/5 gene.

Following are my specific observations

1.       It is not very clear that MoNOT3 and 5 are same gene or not? How much % similarity they showed with each other?

2.        If they are highly similar to each other, from what region authors designed the trasformation construct? Whether they taregeted MoNOT3 or 5.

3.       If they are excatly similar auhtors should study their promoters to clarify their expression in different organs of fungi and their expression pattern.

4.       From the Fig 5 panel a, monot3/5 deletion mutants showed longer hyphal growth compared to WT and complements?

5.       Authors should provide the data of fig 6 a in the form of table or graph along with bnumber of replicates and stats.

6.       Authors should also performed the staining of the hyphal over rice sheath to show the infection structures.

Comments on the Quality of English Language

Please improve typo errors

Author Response

Dear reviewer,

We are grateful for the chance to improve our manuscript with revision. We have fully revised the manuscript. Our responses are highlighted in blue.

We hope this revised manuscript is acceptable for publication in International Journal of Molecular Science. If you need any additional information or recommend any additional changes, please let me know. Once again, we appreciate your effort to improve the quality of our manuscript.

Sincerely,

Sook-Young Park

Reviewer 3

1. It is not very clear that MoNOT3 and 5 are same gene or not? How much % similarity they showed with each other?

-- Sorry for the confusion. The MoNOT3/5 gene is a single gene. To clarify this discrepancy, we conducted further analysis. The MGG_08101 gene was named MoNOT3 based on protein comparison with ScNot3 and ScNot5, as well as the conserved protein in the N-terminal region. All parts of the paper have been revised. In addition, as with Fusarium graminearum, we did not detect the presence of the MoNOT5 gene in the M. oryzae genome.

2. If they are highly similar to each other, from what region authors designed the trasformation construct? Whether they targeted MoNOT3 or 5.

-- Based on the first answer, the second question will be replaced with the first answer.

3. If they are exactly similar auhtors should study their promoters to clarify their expression in different organs of fungi and their expression pattern.

-- Based on the first answer, the third question also will be replaced with the first answer.

4. From the Fig 5 panel a, monot3/5deletion mutants showed longer hyphal growth compared to WT and complements?

-- What the reviewer thinks may be correct. However, as indicated by the arrow in Figure 5a, the mutants appeared to be hooking together to form an appressorium. However, as shown in Figure 5c, only 3% and 8% were able to form an appressorium after 8 h and 24 h, respectively. Instead of interpreting this as extended hyphal growth, it seems more fitting to consider it as a failure of cell development to create an appressorium.

5. Authors should provide the data of fig 6 a in the form of table or graph along with number of replicates and stats.

-- We added a graph for calculation of diseased leaf area.

6 Authors should also have performed the staining of the hyphal over rice sheath to show the infection structures.

-- The reviewer's recommendation is exactly what I agree with. Unfortunately, the student who conducted this research has graduated, so there are no experts in our group available to continue the experiment. However, I believe that the pathogenicity data obtained from rice leaves and onion epidermal cells surpasses the data from rice sheaths, providing a clearer depiction of the situation for wild-types, mutants, and complements.

7. Please improve typo errors

-- We revised.

Round 2

Reviewer 3 Report

Comments and Suggestions for Authors

The authors improved the article and may be accepted.